# Plastid Phylogenomic Analyses Reveal the Taxonomic Position of *Peucedanum franchetii*

**DOI:** 10.3390/plants12010097

**Published:** 2022-12-24

**Authors:** Boni Song, Changkun Liu, Dengfeng Xie, Yulin Xiao, Rongming Tian, Zixuan Li, Songdong Zhou, Xingjin He

**Affiliations:** Key Laboratory of Bio-Resources and Eco-Environment of Ministry of Education, College of Life Sciences, Sichuan University, Chengdu 610065, China

**Keywords:** Apiaceae, *Ligusticopsis*, *Peucedanum franchetii*, new combination, plastome, phylogeny, taxonomy

## Abstract

*Peucedanum franchetii* is a famous folk medicinal plant in China. However, the taxonomy of the *P. franchetii* has not been sufficiently resolved. Due to similar morphological features between *P. franchetii* and *Ligusticopsis* members, the World Flora Online (WFO) Plant List suggested that this species transformed into the genus *Ligusticopsis* and merged with *Ligusticopsis likiangensis*. However, both species are obviously diverse in leaf shape, bracts, and bracteoles. To check the taxonomic position of *P. franchetii*, we newly sequenced and assembled the plastome of *P. franchetii* and compared it with nine other plastomes of the genus *Ligusticopsis*. Ten plastomes were highly conserved and similar in gene order, codon bias, RNA editing sites, IR borders, and SSRs. Nevertheless, 10 mutation hotspot regions (*infA*, *rps8*, *matK*, *ndhF*, *rps15*, *psbA-trnH*, *rps2-rpoC2*, *psbA-trnK*, *ycf2-trnL*, and *ccsA-ndhD*) were still detected. In addition, both phylogenetic analyses based on plastome data and ITS sequences robustly supported that *P. franchetii* was not clustered with members of *Peucedanum* but nested in *Ligusticopsis. P. franchetii* was sister to *L*. *likiangensis* in the ITS topology but clustered with *L. capillacea* in the plastome tree. These findings implied that *P. franchetii* should be transferred to genus *Ligusticopsis* and not merged with *L*. *likiangensis*, but as an independent species, which was further verified by morphological evidences. Therefore, transferring *P. franchetii* under the genus *Ligusticopsis* as an independent species was reasonable, and a new combination was presented.

## 1. Introduction

The genus *Peucedanum*, with *P. officinale* L. as type species [1], is one of the largest taxonomically notorious genera of the Apiaceae family and has a broad circumscription widely distributed in Eurasia and Africa, with Europe and East Asia as the major distribution centers [2,3]. The genus includes about 100–120 species in total, and 40 species (33 species are endemic) occur in China [4,5]. However, the plants of this genus exhibit great diversity in flowers, leaves, bracteoles, and mericarps [3]. Moreover, all previous molecular phylogenetic analysis supported that the genus *Peucedanum* is not monophyletic [5]. Therefore, some authors have implemented several taxonomic changes about this genus [6,7,8,9]. Reduron et al. confirmed that the restitution of *Cervaria*, *Imperatoria*, *Oreoselinum*, *Pteroselinum*, *Thysselinum*, *Xanthoselinum,* and *Holandrea* relied on morphological and phytochemical data [10]. Pimenov et al. suggested all members of *Peucedanum* should be shifted into other genera, excluding 8–10 species in P. sect. *Peucedanum* [11]. Winter et al. [12] transformed 24 *Peucedanum* species into *Afroligusticum* C. Norman, *Cynorhiza* Eckl. & Zeyh., and *Lefebvrea* A. Rich. Although numerous taxonomic treatments for *Peucedanum* members have been achieved, the revision of this genus is still on the way, especially for those taxa endemic to China.

*Peucedanum franchetii* C.Y.Wu & F.T.Pu is endemic to China, growing on grassland or calcareous hillside at an altitude of about 3000 m. Due to dorsally compressed mericarps with slightly prominent dorsal ribs and narrowly winged lateral ribs appearing in *P. franchetii* (Figure 1), this species is deemed to be a member of the genus *Peucedanum* [5]. However, it is similar to the genus *Ligusticopsis,* which was established by Leute in 1969 with the type species *Ligusticopsis rechingeriana* (H.Wolff) Leute [13]. Therefore, in the world flora plant list (http://worldfloraonline.org (accessed on 28 November 2022)), *P. franchetii* was regarded as a synonym of *Ligusticopsis likiangensis* (H.Wolff) Lavrova & Kljuykov. However, making detailed field investigations and collecting the material of this plant in the type locality, we found *P. franchetii* was extraordinarily different from *L. likiangensis* in leaf shape, bracts, and bracteoles. Therefore, we hypothesize that *P. franchetii* may be not the synonym of *L. likiangensis* and could be an independent species. Moreover, the phylogenetic position of *P. franchetii* has not been inferred so far.

In this work, we newly sequenced the complete plastome of *P. franchetii*. Together with nine previously published *Ligusticopsis* plastomes, we carried out a comprehensive comparative analysis among the ten plastomes. Then, we conducted phylogenetic analyses based on plastome data and ITS sequences to confirm the position of *P. franchetii*. Finally, combining with morphological evidence, we made a taxonomic revision for *P. franchetii*.

## 2. Materials and Methods

### 2.1. Taxon Sampling, DNA Extraction, and Sequencing

The fresh leaf of *P. franchetii* used in this experiment was collected from its natural habitat (Figure 1) in Eryuan, Yunnan, China (type locality), and dried in silica gel. The formal identification of *P. franchetii* plant was performed according to the *Flora of China* [5]. The voucher specimens are currently stored at the herbarium of Sichuan University (Chengdu, China) under deposition number LCK2020001. Total genomic DNA was extracted from silica gel dried leaves, employing the modified cetyltrimethylammonium bromide (CTAB) method [14], and carried on the Illumina NovaSeq platform at Personalbio (Shanghai, China), applying the paired-end 150 bp reads with an average insert size of 300–400 bp. Then, the obtained raw data were trimmed by removing adaptors and low-quality reads using AdapterRemoval v2 (trimwindows = 5; minlength = 50) [15], and finally, we filtered low-quality reads to gain high-quality reads with fastP v0.15.0 (-n 10 and -q 15) [16].

In addition, total DNA was also employed to amplify ITS sequence. We operated 30 µL amplification system, including 2 µL extracted total DNA, 10 µL ddH_2_O, 15 µL Taq MasterMix (CWBio, Beijing, China), 1.5 µL of 10 pmol µL^−1^ forward primers (ITS-4: 5′-TCC TCC GCT TAT TGA TAT GC-3′), and 1.5 µL of 10 pmol µL^−1^ reverse primers (ITS-5: 5′-GGA AGT AAA AGT CGT AAC AAG G-3′) [17]. PCR cycling profile consisted of an initial denaturation step at 94 °C for 3 min, followed by denaturation step at 94 °C for 45 s, 30 cycles of 45 s at 94 °C, annealing at 55 °C for 45 s and extension at 72 °C for 45 s, a final extension for 7 min at 72 °C, and storage at 4 °C. All PCR products were separated, manipulating a 1.5% (wv^−1^) agarose TAE gel and delegated Sangon Biotech (Shanghai, China) for sequencing. DNAstar–SeqMan [18] was occupied to edit the newly sequenced DNA and gain consensus sequences.

### 2.2. Plastome Assembly and Annotation

The plastome of *P. franchetii* was de novo assembled via GetOrganelle v1.6.2 software [19] considering the plastome of *Ligusticopsis brachyloba* (Franch.) Leute (NC_065500) as reference. To assess the sequencing depth and overlapping contigs, the cleaned reads were mapped to the reference plastome by Geneious v9.0.2 [20]. The Plastid Genome Annotator (PGA) was applied to annotate the plastome [21], followed by the validation of annotation adopting online GeSeq tool [22] with default parameters. The plastome of *L. brachyloba* (NC_065500) was also chosen as the reference to predict protein-coding genes (PCGs) and transfer RNA (tRNA) genes and ribosomal RNA (rRNA) genes. Finally, we checked the annotation results with BLAST and DOGMA [23] and corrected the wrong annotation in Geneious v9.0.2 [20] manually. A detailed plastome map was drawn by the online Organellar Genome DRAW tool (OGDRAW: https://chlorobox.mpimp-golm.mpg.de/geseq.html, accessed on 1 October 2022) [24]. The newly yielded ITS sequence and plastome of *P. franchetii* were submitted into the NCBI under the accession numbers OP740295 and OP745624, respectively. Additionally, we downloaded 54 ITS sequences and 54 plastomes of Apioideae for the phylogenetic analyses. The names of the species and GenBank accession numbers for all samples were listed in Appendix A.

### 2.3. Codon Usage, RNA Editing, and SSRs Analyses

To identify codon usage pattern and relative synonymous codon usage (RSCU) values [25], CodonW v1.4.2 program [26] was used to analyze the codon usage bias, and the heatmap was depicted by TBtools [27]. To acquire the composition and characteristics of RNA editing, 41 shared protein-coding genes for 10 plastomes were carried out through the online PREP-Cp program with a cutoff value of 0.8, and other parameters were default to predict RNA editing sites [28]. Simple sequence repeats (SSRs) of ten plastomes were calculated by MISA web (http://misaweb.ipk-gatersleben.de/, accessed on 2 October 2022) [29] tool with the following criteria: 10, 5, 4, 3, 3, and 3 repeat units for mono-, di-, tri-, tetra-, penta-, and hexa-nucleotides, respectively.

### 2.4. Comparative Plastome Analyses

The expansion or contraction of the IR regions may lead to the size variation of the plastome; the boundaries of IR regions in ten plastomes were analyzed by IRscope [30] and manually inspected in Geneious v9.0.2 [20]. To better determine whether any specific pattern of structural variation exists in the ten plastomes, gene arrangement comparison was visualized by the Mauve alignment program [31] with default settings in Geneious v9.0.2 [20]. Furthermore, the sequence identity among the ten species was detected using mVISTA software (https://genome.lbl.gov/vista/mvista/submit.shtml, accessed on 3 October 2022) [32] with the default settings, and *P. franchetii* was chosen as the reference.

### 2.5. Identification of Divergence Hotspots

To further discern the mutation hotspot regions, the protein-coding regions, non-coding regions, and intron regions of the ten plastomes were first extracted in Geneious v9.0.2 [20] and aligned employing MAFFT version 7.427 [33] with default parameters. Then, the alignments were manually adjusted. Finally, the nucleotide diversity (Pi) was calculated for alignments with more than 200 bp in length carrying DnaSP version 5.0 [34].

### 2.6. Phylogenetic Analyses

To pinpoint the phylogenetic position of *P. franchetii*, 54 ITS and 54 complete plastome sequences (Appendix A) were employed to reconstruct the phylogenetic trees. *Anthriscus sylvestris* (L.) Hoffm and *Anthriscus cerefolium* (L.) Hoffm were selected as outgroups according to the results of Wen et al. [35]. Then, 54 ITS and 54 plastome sequences were respectively aligned in Geneious prime 2021.1.1 plugin MAFFT 7.427 [33] and modified manually. Alignments were used for maximum likelihood analyses (ML) and Bayesian inference (BI) methods. The best-scoring ML tree was inferred adopting RAxML version 8.2.11 [36] with the GTRGAMMAX model for each partition, and branch support was assessed by the rapid bootstrap algorithm [37] with 1000 replicates. For BI analysis, four parallel Markov chain Monte Carlo (MCMC) runs were executed in MrBayes version 3.2.7 [38] and the best-fitting substitution model (SYM + I + G) for ITS dataset and (GTR + I + G) for plastome dataset determined by Modeltest v3.7 [39]. A total of 1,000,000 generations were run with sampling every 500 generations, and the first 25% of samples were discarded as burn-in and the remainder were maintained to produce the consensus tree. Results of phylogenetic analyses were visualized and exhibited in FigTree v1.4.2 software [40] and beautified by the online tool iTOL (https://itol.embl.de/itol.cgi, accessed on 4 October 2022) [41].

## 3. Results

### 3.1. Characteristics of Plastome

The length of the complete plastome sequence in *P. franchetii* was 148,281 bp and displayed a typical quadripartite structure (Figure 2), containing one LSC region (92,298 bp) and one SSC region (17,691 bp) separated by a pair of IR regions (19,146 bp). Moreover, the total GC content was 37.5%. The plastome encoded 113 unique genes, and these genes had four categories: self-replication, genes for photosynthesis, other genes, and genes of unknown function. Among them, 15 genes harbored a single intron, and 3 genes (*ycf3*, *clpP*, and *rps12*) had 2 introns (Appendix A).

The study also compared *P. franchetii* plastome with nine *Ligusticopsis* plastomes. For the ten plastomes, the overall size ranged from 147,752 bp (*L. involucrata*) to 148,633 bp (*L. brachyloba*). They hold the same unique gene number, comprising 79 protein-coding genes, 30 tRNA genes, and 4 rRNA genes. The overall GC content was corresponding in the ten plastomes (37.3–37.5%). Hence, the structure of the plastome appeared to be largely conserved across the ten species (Table 1).

### 3.2. Codon Usage, RNA Editing in Protein-Coding Genes

Our study summarized the codon usage frequency of 79 protein-coding genes among the 10 plastomes. The result demonstrated that codon usage bias was similar in them. The 79 protein-coding genes encoded 22,632–22,553 codons. In these codons, the Leu was encoded by the highest number of codons 2399–2410 (10.64%–10.65%), while the Cys was the least 231–237 (1.02–1.05%) (Appendix A). Figure 3 showed the relative synonymous codon usage (RSCU) values of all protein-coding genes for the ten plastomes. The RSCU values ranged from 0.34 to 2.00. The UAA codon possessed the highest value (RSCU = 2.0) in the *P. franchetii*, *L. brachyloba*, *L. capillacea,* and *L. scapiformis* plastomes, while the AGC codon possessed the smallest value (RSCU = 0.34) only in the *L. involucrata* plastome. The RSCU value of codons AUG and UGG was 1.00 (Appendix A). In addition, the RSCU values of 30 codons were greater than 1.00, and these codons were ended with A/U, except UUG in ten plastomes.

This work also predicted the RNA editing sites of the ten plastomes. We found 22–23 protein-coding genes and identified 57–59 potential RNA editing sites in the 10 plastomes (Appendix A, Figure 4). Among these protein-coding genes, the *ndh* gene had the most editing sites (22), while other genes (*atpA*, *ccsA*, *petG*, *psaI*, *psbE*, *psbF*, *rpl20*, *rpoA*, and *rps2*) only had one editing site. Among all detected RNA editing sites, the second codon position occupied the most editing sites (43–45), followed by the first codon position (14), but no sites detected in the third codon position (Figure 4A). Although ten plastomes appeared to have a similar pattern of RNA editing, one specific editing site had been picked out: *petD* (one site; only identified in *L. involucrata*) (Figure 4B). Moreover, the amino acid conversion S to L occurred most frequently, while A to V and R to C occurred least (Appendix A).

### 3.3. Simple Sequence Repeats (SSRs)

We estimated and compared the SSRs in the ten plastomes. The total number of SSRs altered from 71 to 84, of which the LSC region had the richest repeats (46–62) and the SSC region (9–14) had the poorest following IR regions (8–14) (Appendix A, Figure 5A). Figure 5B displayed the number of different repeat types in ten plastomes. Among these SSRs, the mononucleotide repeats were the maximum (36–42), followed by the dinucleotides (18–24) and the hexanucleotide repeats, which were the minimal (0–1).

### 3.4. Plastome Comparison and Hotspots Identification

In the current study, we conducted a detailed comparison of the IR boundaries among the ten plastomes. The LSC/IRa junction was located in the *ycf2* gene and expanded 576 bp into the *ycf2* gene except for *L. brachyloba* (585 bp). The SSC/IRa junction was located in the intergenic spacer region between the *ndhF* and *ycf1* gene, but expanded 72 bp into *ndhF* gene in *L. capillacea*. The SSC/IRb junction was located in the *ycf1* gene and expanded 1984–2029 bp into the *ycf1* gene. The LSC/IRb junction was located between the *trnL* and *trnH* genes, and the *trnL* gene was 1860–2771 bp away from LSC, while the *trnH* gene had the same length (6 bp) away from the IRb borders (Figure 6).

The Mauve result demonstrated that the gene arrangement was highly conservative and, moreover, the gene retained a similar sequence among the ten plastomes (Figure 7 and Figure 8).

The nucleotide diversity (Pi) was calculated to analyze the sequence divergence level of four regions across the ten plastomes. The results indicated that the divergence level of LSC and SSC regions were higher than the IR regions (Figure 9). We also screened ten mutation hotspot regions as candidate DNA barcodes, including five protein-coding genes (*rps8*, *matK*, *infA*, *ndhF*, and *rps15*) with the Pi > 0.00344 (Figure 9A) and five non-coding genes (*psbA-trnH*, *rps2-rpoC2*, *psbA-trnK*, *ycf2-trnL*, and *ccsA-ndhD*) with the Pi > 0.01950 (Figure 9B).

### 3.5. Phylogenetic Analyses

We used 54 ITS sequences and 54 plastome sequences of Apioideae to conduct the phylogenetic analyses (Appendix A). Although the topological tree between the ITS and plastome were conflicting, both demonstrate that *P. franchetii* was nested in the genus *Ligusticopsis*. In the plastome tree, *P. franchetii* clustered with *L. capillacea* and resolved as a sister to other *Ligusticopsis* taxa with high support (BS = 100%, PP = 1.00) (Figure 10A). However, in the ITS topology, *P. franchetii* clustered with *L. likiangensis* also with high support (BS = 92%, PP = 1.0) and resolved it as a sister to other *Ligusticopsis* taxa (Figure 10B). Moreover, both phylogenetic trees also suggested that *Ligusticum daucoides* (Franch.) Franch and *Ligusticum oliverianum* (de Boiss.) Shan are nested in the genus *Ligusticopsis* (Figure 10).

## 4. Discussion

### 4.1. Plastome Characteristics

We reported the newly sequenced and assembled complete plastome of *P. franchetii* and performed a comparative analysis with nine other *Ligusticopsis* species. The results indicated that ten plastomes exhibited a typical quadripartite structure, which was in accordance with the plastome structure of Apiaceae published previously [42,43,44,45,46,47,48]. Furthermore, all plastomes were similar in genome size, gene order, and GC content. These results demonstrated that ten plastomes were highly conserved. In addition, higher GC content was detected in IR regions compared to the LSC and SSC regions, for which was present four rRNA (*rrn23*, *rrn16*, *rrn5*, and *rrn4.5*) genes with high GC content (50–56.4%) in IR regions [49,50].

### 4.2. Codon Usage Analyses

Codons act as a bridge between nucleic acids and proteins in spreading genetic information [51]. Each amino acid has two or more codons and each has its preferred codon for codon usage bias (CUB). CUB is a common phenomenon that widely exists in organisms [52,53], and it has been hypothesized that it controls translation dynamics such as reliability, accuracy, and protein folding [54]. Recent research has proclaimed that codon usage greatly influences the evolution of the plastome [55,56]. In this investigation, Leu was the more frequent amino acid and Cys was the least. The effects were consistent with those observed in other plant plastome studies [57,58,59,60,61]. The most preferred codons end with A/U (Appendix A, Figure 3), which was commonly examined in previously reported genera of flowering plants [62,63,64]. We also counted RSCU values for 64 mutant codons, 30 of which had RSCU values larger than 1.00, and codons AUG and UGG had no bias (RSCU = 1.00). These results correspond to prior plastome studies, demonstrating that usage bias of particular codons was induced by adaptation evolution of the plastome [62].

### 4.3. RNA Editing Analyses

RNA editing as a post-transcriptional modification process is important for correcting DNA mutations on the RNA level [65,66,67]. In the present study, the editing sites were cytidine (C) to uridine (U) conversions in ten plastomes (Appendix A), which was reported in higher plants [43,44,45]. Among these RNA editing sites, changing a nucleotide at the second position was more common than other position shifts. It corresponded to the general characteristics of plastid gene RNA editing in higher plants [59,68,69,70]. Moreover, a multitudinous number was discovered in the *ndhB* gene, which was line with the previous conclusion that *ndh* group genes have the largest number of plastome editing sites in flowering plants [70].

### 4.4. Simple Sequence Repeats (SSRs) Analyses

SSRs are usually small tandem single mononucleotide repeats, which show intraspecific repeat number differences [71,72]. Owing to the characteristics of SSR’s codominant inheritance, high reproducibility, multiallelic composition, richness, and ease of detection, it is widely discussed in plant diversity analysis [73,74]. This work gained a total number of (71–84) SSRs, which had coincident results among other Apiaceae in numbers [44,75]. The results also showed that the SSR loci of the LSC region appeared more usually than in the SSC and IR regions; it may be hypothesized that it was relevant to the longer LSC region [43]. Moreover, the mononucleotide repeats were abundant and mostly composed of A/T motifs. The same results were reported in many other angiosperms plastomes [76,77]. However, two hexanucleotide simple sequence repeats (AATATT/AATATT) were observed in the plastome of *L. capillacea*, though not in other plastomes, which could be served as specific molecular markers to recognize this species. Such studies implied that SSRs changed the plastome and played an important role in detecting genomic variability [58].

### 4.5. The IR/SC Boundaries of the Ten Plastomes

IR contraction and expansion are extremely common in plastomes of angiosperm plants [78,79,80,81]. Here, we compared the IR/SC boundaries and found that the crossroads and surrounding genes of four sections IRa/SSC/IRb/LSC were the same. Moreover, the ten plastomes had same gap between the IRb/LSC junctions. These results further justified the conservation of the ten plastomes. Our research supported this hypothesis, demonstrating that the IR region is more conserved and that most substitutions occur in the SSC and LSC regions. The same phenomena have been achieved in other plastid genomes as well [82,83,84,85].

### 4.6. Candidate DNA Barcodes

DNA barcodes are considered as short DNA sequences with adequate variations to identify species [86]. In plants, three variable loci (*matK*, *rbcL*, and *trnH-psbA*) from plastome were extensively defined as the standard DNA barcodes. But these fragments had been proved to be insufficient for the identification of a taxonomically complex group [87]. For example, Liu et al. operated the *rbcL* gene as DNA barcodes suffered from weak solutions among *Peucedanum* [43]. Therefore, additional loci are urgently needed to be developed for specific taxa. In this study, we screened the ten most divergent loci, containing five protein-coding genes (*rps8*, *matK*, *infA*, *ndhF*, and *rps15*) and five non-coding protein genes (*psbA-trnH*, *rps2-rpoC2*, *psbA-trnK*, *ycf2-trnL*, and *ccsA-ndhD*) based on the complete plastome. These regions could be served as potential DNA barcodes for species identification in *Ligusticopsis*.

### 4.7. Phylogenetic Relationships and Taxonomic Implication of P. franchetii

Although plastome is highly conserved in genome structure, gene order, and content, it shows highly variable characters. Therefore, plastome data have been applied to generate highly resolved phylogeny in plant species, especially in taxonomically complex groups [88,89,90]. In this article, we constructed the phylogenetic framework depending on plastome data to make a definite position of *P. franchetii.* Furthermore, we also performed phylogenetic analysis using ITS sequences, mainly due to the fact that chloroplast genomes only represent maternal inheritance.

Previous phylogenetic analysis based on nrDNA ITS sequences performed by Zhou et al. [91] confirmed that the generitype species *Peucedanum officinale* was distant from *Ligusticopsis* (including the type species *Ligusticopsis rechingeriana*). Although there was little difference between the plastome and ITS topologies, the *P. franchetii* fell into genus *Ligusticopsis* with high support value, which suggested that *P. franchetii* was a member of genus *Ligusticopsis.* The morphological characteristics of *P. franchetii* also supported the above phylogenetic results, such as pinnate leaves, pinnate and linear coexisted bracteoles, conspicuous calyx teeth, and numerous vittae in mericarp commissure and each furrow (Figure 1, Appendix A). All these morphological traits were in accordance with the genus *Ligusticopsis.* Thus, *P. franchetii* should be transferred from genus *Peucedanum* to *Ligusticopsis.*

*P. franchetii* was regarded as a synonym of *L. likiangensis* in the World Flora plant list (http://worldfloraonline.org, accessed on 10 December 2022). However, our results did not support this treatment. Although *P. franchetii* was a sister to *L. likiangensis* in the ITS topology, this species clustered with *L. capilliacea* in the plastome tree. Moreover, obviously different morphological characters were detected for the both species. The absent bracts and pinnate and linear coexisted bracteoles were observed in *P. franchetii*, while pinnate and linear coexisting bracts and pinnate bracteoles were found in *L. likiangensis* (Appendix A). Therefore, merging *P. franchetii* into *L. likiangensis* was inappropriatem and *P. franchetii* should be regarded as a dependent species of genus *Ligusticopsis*.

In addition, our phylogenetic tree also indicated that *Ligusticum daucoides* and *Ligusticum oliverianum* nested in the genus *Ligusticopsis* and belonged to Selineae. Both species were obviously distant from the type species of *Ligusticum* (*Ligusticum scoticum* L.), which is located in *Acronema* Clade [91,92]. These results supported the taxonomic treatments performed by Pimenov [93], who transferred both species into the genus *Ligusticopsis*. This conclusion is consistent with the definition of *Ligusticopsis* based on the chloroplast genome and morphological boundary performed by Li et al. [44]. Our results with high supports provided additional evidence to accept the previous treatments.

### 4.8. Taxonomic Treatment

*Ligusticopsis franchetii* (C.Y.Wu & F.T.Pu) B.N.Song, C.K.Liu & X.J.He, comb. nov.

≡ *Peucedanum franchetii* C.Y.Wu & F.T.Pu in Wang Wentsai (ed.), Vasc. Pl. Hengduan Mount. 1349 (1993)

Type: China. Yunnan: in pratis calcareis montis Hee-can-men, 3000 m, 3 October 1884, Delavay 192 (isolectotype: P! P02272009; lectype P! P02272008) [93].

Descriptions: Perennial herbs, 20–30cm, pallid green, often purplish-tinged. Stems several, hollow, puberulous above. Leaf blade long ovate, pinnate, thinly coriaceous, abaxially strongly reticulate, white villous, margins dentate and slightly reflexed; pinnae 1–2 × 0.5–1 cm, 2–3 pairs, lateral pinnae rhombic or oblique ovate, base cuneate or truncate, apical pinnae ovate, base cuneate, decurrent. Umbels terminal, 2–3 cm across; peduncles elongate, straight, apex villous; bracts absent; rays 8–14, 1–2 cm, 4-angled, inner faces white hispid, outer faces glabrous; bracteoles 8–10, pinnate and linear coexisted. Umbellules 12–16-flowered. Calyx teeth conspicuous. Petals white. Styles longer than stylopodium. Fruit ovoid-oblong, ca. 3 × 2 mm, glabrous; dorsal ribs prominent, lateral ribs winged, wings ca. 1 mm, ca. 1/3 width of body, thin; vittae 1–3 in each furrow, 4–8 on commissure.

Distribution and habitat in China: *Ligusticopsis franchetii* is endemic to south-western China (north-western Yunnan). It grows in alpine meadows in limestone areas, at an elevation of 2900–4000 m.

Additional specimens examined: China—Yunnan: Lijiang County, 2900 m, 4 September 1992, N.H. Wang and C.N. Zhang 92027 (NAS); Dali, 3037 m, 22 August 2021, Z.X. Li 210822-2-1 (SZ); 30 October 2021, C.K. Liu LCK20211030001 (SZ).

## 5. Conclusions

To summarize, comparative analyses displayed that the genome structure and size, number, gene order, GC content, RNA editing, and codon usage were extremely conservative among the ten plastomes. However, ten highly variable regions (*rps8*, *matK*, *infA*, *ndhF*, *rps15, psbA-trnH*, *rps2-rpoC2*, *psbA-trnK*, *ycf2-trnL*, and *ccsA-ndhD*) were identified, which may be strong potential DNA barcodes for species identification and phylogenetic relationship construction. Furthermore, both morphological characteristics and molecular data robustly supported that *P. franchetii* should be transferred from the genus *Peucedanum* to *Ligusticopsis* as an independent species. In a word, our result will be beneficial to future phylogeny, taxonomy, and evolutionary studies of the genus *Ligusticopsis.*

## Figures and Tables

**Figure 1 plants-12-00097-f001:**
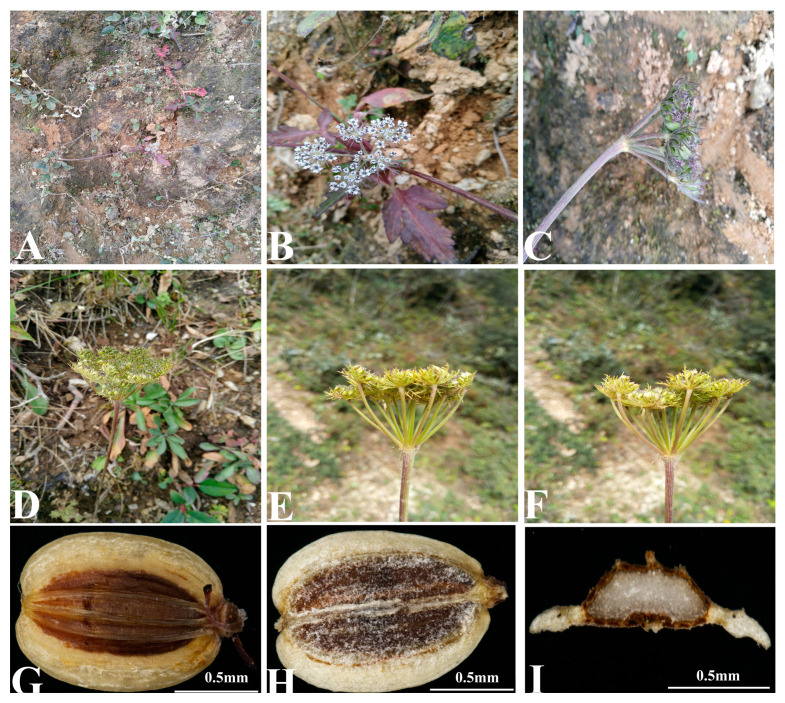
Illustrations of *Peucedanum franchetii*. (**A**) Habit; (**B**) basal leaves; (**C**) stem leaves; (**D**) umbel with bracts and bracteoles; (**E**) flower; (**F**) young mericarp with calyx teeth; (**G**) dorsal of mericarp; (**H**) commissural of mericarp; (**I**) cross-section of mericarp. Scale bars: 0.5 mm (**G**,**H**,**I**).

**Figure 2 plants-12-00097-f002:**
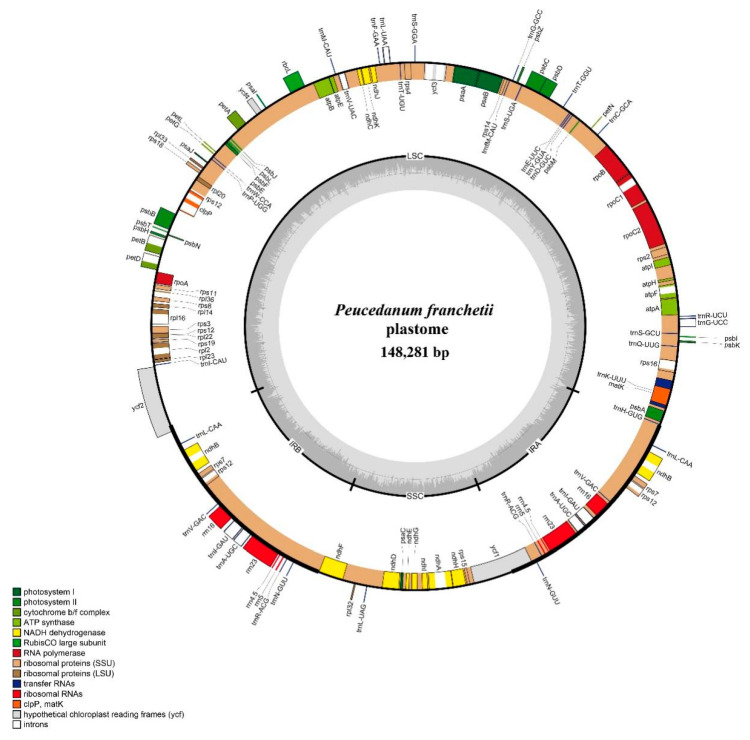
Gene map of the *P. franchetii* plastome. The genes exhibited outside of the circle are transcribed clockwise, while those are counterclockwise inside. The genes belonging to different functional groups are color-coded.

**Figure 3 plants-12-00097-f003:**
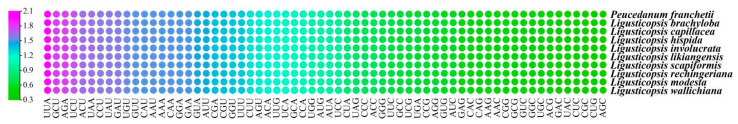
The RSCU values of all concatenated protein-coding genes for ten plastomes. Color key: the purple values represent higher RSCU values while the green values indicate lower RSCU values.

**Figure 4 plants-12-00097-f004:**
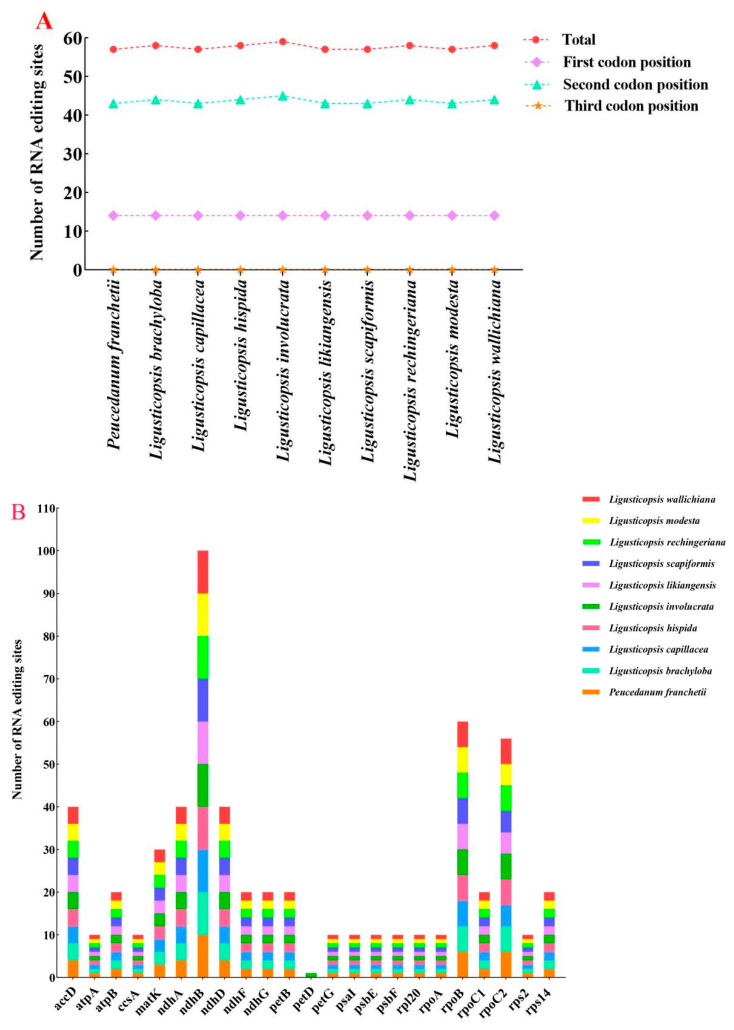
The RNA editing sites in ten plastomes. (**A**) Numbers of RNA editing sites distributed in different codon positions; (**B**) numbers of RNA editing sites presented in genes.

**Figure 5 plants-12-00097-f005:**
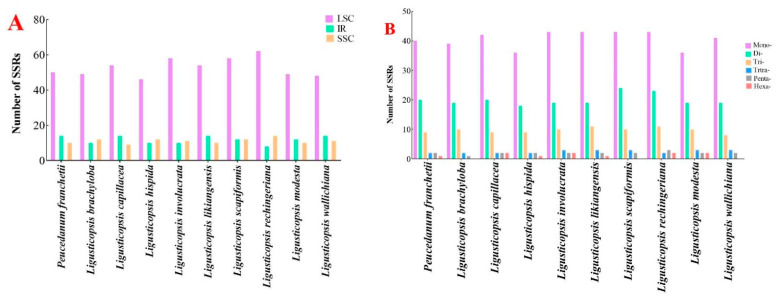
Analyses of simple sequence repeats (SSRs) in ten plastomes. (**A**) Presence of SSRs in LSC, SSC and IR; (**B**) numbers of different repeat types.

**Figure 6 plants-12-00097-f006:**
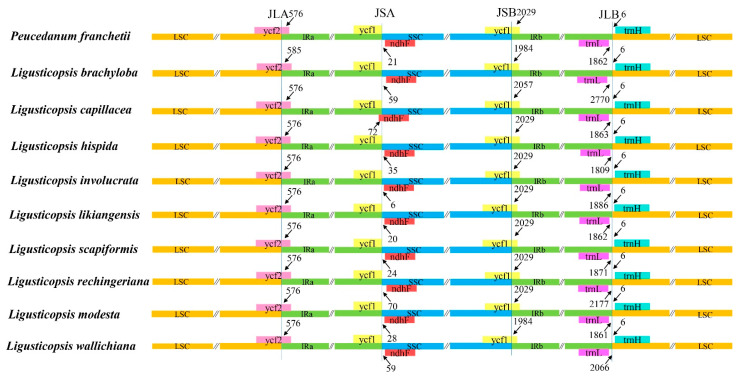
Comparison of the border positions of LSC, SSC, and IR regions across the ten plastomes. Functional genes and truncated fragments are denoted by colored boxes. The sizes of gene fragments located at boundaries are indicated by the base pair lengths.

**Figure 7 plants-12-00097-f007:**
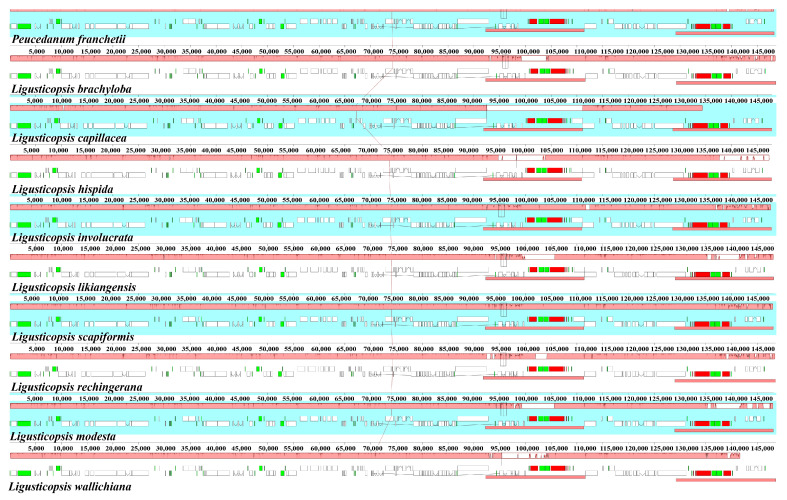
Mauve alignment of ten plastomes. Local collinear blocks within each alignment are represented by blocks of the same color connected with lines.

**Figure 8 plants-12-00097-f008:**
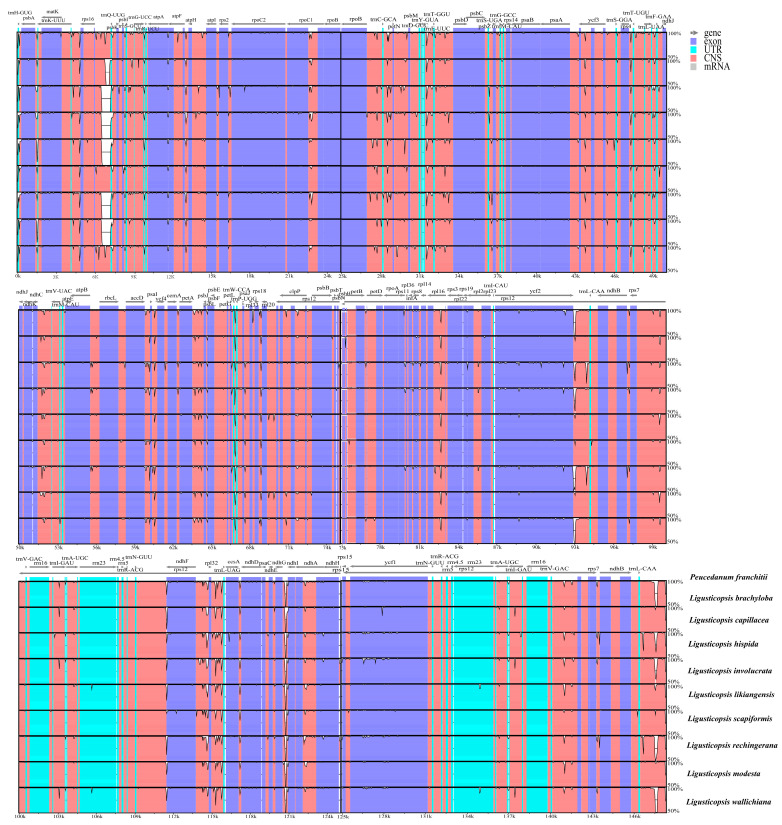
Sequence identity plot comparing the ten plastomes using mVISTA. The y-axis corresponds to percentage identity (50–100%), while the x-axis shows the position of each region within the locus. Arrows indicate the transcription of annotated genes in the reference genome.

**Figure 9 plants-12-00097-f009:**
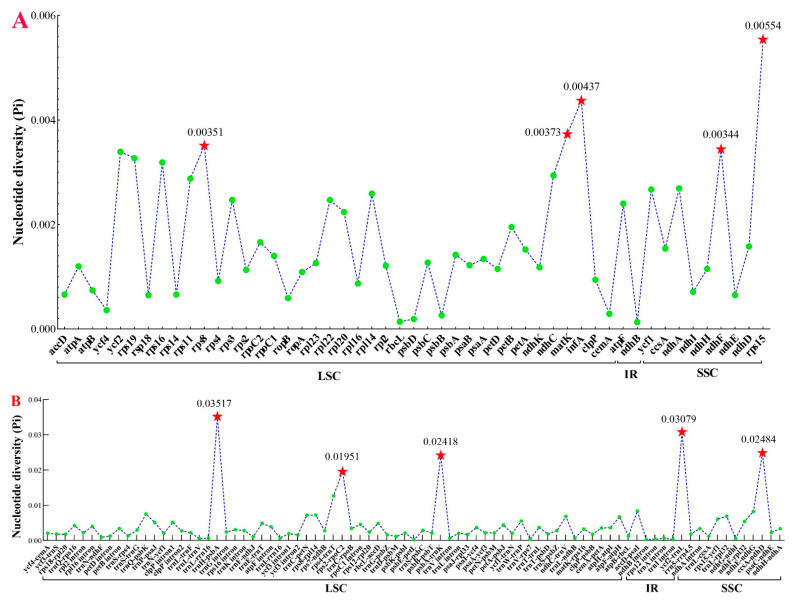
Comparative analysis of the nucleotide diversity (Pi) values among the ten plastomes. (**A**) protein-coding genes; (**B**) non-coding and intron regions.

**Figure 10 plants-12-00097-f010:**
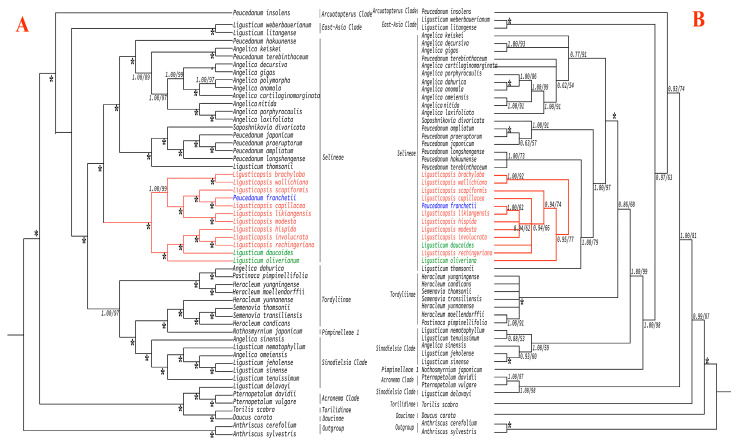
Phylogeny of the 54 taxa inferred from maximum likelihood (ML) and Bayesian inference (BI) analyses. Branch support was assessed using ML bootstrap percentage (BP) and Bayesian posterior probability (PP), and internal branches with less than 100 BP/1.0 PP were indicated with corresponding values. (**A**) Phylogenetic tree of 54 Apioideae taxa based on plastomes. (**B**) Phylogenetic tree of 54 Apioideae taxa based on ITS sequences. (* means the node with BS = 100 and PP = 1.00).

**Table 1 plants-12-00097-t001:** Comparative analyses of ten plastomes. *P. franchetii* included.

Taxon	Length (bp)					Number of Genes (Unique)			
	Genome	LSC	SSC	IR	Total	CDS	rRNA	tRNA	GC (%)
*P. Franchetii* C.Y.Wu & F.T.Pu	148,281	92,298	17,691	19,146	113	79	4	30	37.50%
*L. brachyloba* (Franch.) Leute	148,633	92,265	17,588	19,390	113	79	4	30	37.40%
*L. capillacea* (H.Wolff) Leute	147,808	91,907	17,503	19,199	113	79	4	30	37.50%
*L. hispida* Lavrova & Kljuykov	147,797	91,846	17,627	19,162	113	79	4	30	37.40%
*L. involucrata* Lavrova	147,752	91,782	17,560	19,205	113	79	4	30	37.40%
*L. likiangensis* (H.Wolff) Lavrova & Kljuykov	148,196	92,305	17,575	19,158	113	79	4	30	37.50%
*L. scapiformis* (H.Wolff) Leute	148,107	92,214	17,581	19,156	113	79	4	30	37.50%
*L. rechingeriana* Leute	148,525	91,813	17,654	19,529	113	79	4	30	37.30%
*L. modesta* (Diels) Leute	148,133	92,247	17,568	19,159	113	79	4	30	37.50%
*L. wallichiana* Fedde ex H.Wolff	148,594	92,281	17,567	19,373	113	79	4	30	37.40%

## Data Availability

The newly yielded ITS sequence and plastome of *P. franchetii* was submitted into the NCBI with accession numbers OP740295 and OP745624, respectively.

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
