# Peer review of "Plastid Phylogenomic Analyses Reveal the Taxonomic Position of Peucedanum franchetii"

_plants, 2022, doi:10.3390/plants12010097_

Round 1
Reviewer 1 Report (Previous Reviewer 2)
You must to see worldfloraplantlist and discuss the treatment given to P. franchetii

Author Response
Author's Reply to the Review Report
Reviewers' comments:
You must to see worldfloraplantlist and discuss the treatment given to P. franchetii.
Response: Thanks very much for your comments, we have carefully see world flora plant list and added this important information in Introduction section (Line 51-55) and Discussion section (Line 362-368) in our revised version.
Thank you very much for your comments and suggestions.
If there are other errors or further requests, please contact me by e-mail.
Sincerely
Bo-Ni Song
E-mail address: [email protected].
Corresponding authors:
Xin-Jin He; E-mail address: [email protected].

Reviewer 2 Report (Previous Reviewer 1)
The manuscript was much improved with respect to a previous version, and many of the criticisms I raised were addressed.
However there are still two open main problems and a number of minor corrections, as you will find in the annotated manuscript.
Concerning the two main problems, in short:
1) the lack of the type species Peucedanum officinale in the phylogenetic trees (or at least appropriate discussion about the phylogenetic position of this species with respect to Ligusticopsis
2) the formal nomenclatural combination proposed is still contrary to the International Code of Nomenclature for Algae, Fungi and Plants (Art. 41.5)
Author Response
Author's Reply to the Review Report
Reviewers' comments:
The manuscript was much improved with respect to a previous version, and many of the criticisms I raised were addressed. However there are still two open main problems and a number of minor corrections, as you will find in the annotated manuscript. Concerning the two main problems, in short:
Response: Thanks for your positive comments and suggestions, we have addressed the two main problems and carefully corrected a number of minor flaw in our revised version.
1) the lack of the type species Peucedanum officinale in the phylogenetic trees (or at least appropriate discussion about the phylogenetic position of this species with respect to Ligusticopsis.
Response: Thanks for your comments. We have discussed the phylogenetic position of the type species Peucedanum officinale with respect to Ligusticopsis and added this important information in Discussion section (Line 351-353) in our revised version.
2) the formal nomenclatural combination proposed is still contrary to the International Code of Nomenclature for Algae, Fungi and Plants (Art. 41.5)
Response: Thanks for your comments, we are so sorry that we write the informal information. We have corrected it in our revised version according to the International Code of Nomenclature for Algae, Fungi and Plants (Art. 41.5).(Line 386-387)
Thank you very much for your comments and suggestions.
If there are other errors or further requests, please contact me by e-mail.
Sincerely
Bo-Ni Song
E-mail address: [email protected].
Corresponding authors:
Xin-Jin He; E-mail address: [email protected].

Round 2
Reviewer 2 Report (Previous Reviewer 1)
The formal aspects of the new combination are now ok, BUT in this latter version a new relevant problem emerges: what about the synonymy with L. likiangensis, a species that you did not investigate? IF this synonymy is confirmed, then the priority species name is indeed L. likiangensis, which is based on Pleurospermum likiangensis H.Wolff, described in 1929, while Peucedanum franchetii was described much later. Please also note that WFO consider also L. integrifolia as a synonym of L. likiangensis!!
But to which of the two units of diversity the name L. likiangensis applies?
If this name applies indeed to what the authors name L. integrifolia in their manuscript, they should use the former name instead of the latter all across the manuscript.
If this name applies to that the authors name Peucedanum franchetii, the authors can easily understand that they should deeply change all the manusript beginning from the title, so that the main aim would be to check the debated synonymy between L. likiangensis and L. integrifolia. Moreover, this second approach would be more critical, since they did not study topotypical populations of Pleurospermum likiangense but of Peucedanum franchetii.
In any case, the authors must carefully consider the name L. likiangense (also looking at its nomenclatural type if designated), before reaching any taxonomic conclusion and proposing new combinations.
Minor points:
"as a synonym", not "as synonymy"
"H.Wolff", not "H. Wolff" (several times)
"Ligusticopsis likiangensis", not "the Ligusticopsis likiangensis"
Author Response
Author's Reply to the Review Report
Reviewers' comments:
The formal aspects of the new combination are now ok, BUT in this latter version a new relevant problem emerges: what about the synonymy with L. likiangensis, a species that you did not investigate? IF this synonymy is confirmed, then the priority species name is indeed L. likiangensis, which is based on Pleurospermum likiangensis H.Wolff, described in 1929, while Peucedanum franchetii was described much later. Please also note that WFO consider also L. integrifolia as a synonym of L. likiangensis!!
Response: Thanks for your positive comments and suggestions. The reason why we use L. integrifolia in our previous version is that to commemorate the scholar (H.Wolff) Leute who established the genus Ligusticopsis, while ignoring the priority principle of naming law. We are so sorry for this inaccurate statement caused by our inadequate consideration. We also carefully referred the WFO Plant List and revised the manuscript. In our study, L. integrifolia is as a synonym of L. likiangensis and we have corrected L. integrifolia into L. likiangensis in the whole text and the revised portions are tracked with blue words in the text.
But to which of the two units of diversity the name L. likiangensis applies?
If this name applies indeed to what the authors name L. integrifolia in their manuscript, they should use the former name instead of the latter all across the manuscript.
If this name applies to that the authors name Peucedanum franchetii, the authors can easily understand that they should deeply change all the manuscript beginning from the title, so that the main aim would be to check the debated synonymy between L. likiangensis and L. integrifolia.
Moreover, this second approach would be more critical, since they did not study topotypical populations of Pleurospermum likiangense but of Peucedanum franchetii.
In any case, the authors must carefully consider the name L. likiangense (also looking at its nomenclatural type if designated), before reaching any taxonomic conclusion and proposing new combinations.
Response: Thanks for your comments. In our study, L. likiangensis is indeed to the name L. integrifolia and we had amended L. integrifolia as L. likiangensis in whole text.
Minor points:
Thanks for your positive comments and suggestions, we have carefully checked whole text and addressed the minor points in our revised version.
"as a synonym", not "as synonymy"
Response: Thanks for your comments. We have corrected "as synonymy" to "as a synonym".
"H.Wolff", not "H. Wolff" (several times)
Response: Thanks for your comments. We have corrected "H. Wolff" to "H.Wolff".
"Ligusticopsis likiangensis", not "the Ligusticopsis likiangensis"
Response: Thanks for your comments. We have corrected "the Ligusticopsis likiangensis" to "Ligusticopsis likiangensis".
Thank you very much for your comments and suggestions.
If there are other errors or further requests, please contact me by e-mail.
Sincerely
Bo-Ni Song
E-mail address: [email protected].
Corresponding authors:
Xin-Jin He; E-mail address: [email protected].

Round 3
Reviewer 2 Report (Previous Reviewer 1)
The manuscript can now be accepted.
I spotted these minor problems in the abstract: "suggested to transfer", not "suggested that transformed"; "capillacea", not "capilliacea"
This manuscript is a resubmission of an earlier submission. The following is a list of the peer review reports and author responses from that submission.
Round 1
Reviewer 1 Report
The manuscript covers an interesting topic, but it shows several serious flaws which certainly need to be carefully addressed before further consideration for publication in Plants or elsewhere.
1) the stated focus of the paper would concern taxonomy, but there are a number of problems about it. For instance, the authors did not clarify which are the generitype species of the involved genera (Peucedanum, Ligusticopsis, Ligusticum) and whether these species have been included in their study or not. This is crucial.
2) the manuscript is too much detailed concerning the description of the sequenced plastome. Moreover, it is not clear at all why the authors decided to compare the newly produced plastome with those of 10 other Ligusticopsis species. Why only Ligusticopsis, if there are doubts about the generic placement of Peucedanum franchetii? And why the need of this paper, if on the contrary there was already enough confidence about the generic placement of this species?
3) the strictly nomenclatural part lacks of some important information, as for instance reference to the valid publication of the basionym or a clear reference to the literature in wich the name Peucedanum franchetii was lectotypified.
4) the quality of the images is very poor in general. This problem is particularly evident in Figure 10, where I was not even able to read scientific names in the two phylogenetic trees reported.
5) English language needs an extensive revision, since in many parts awkward and non-scientific/precise terms/sentences are reported.
I also directly annotated the manuscript.

Reviewer 2 Report
Figure 1 has no legend, and the quality of A, B and C is very bad.
In figures 8, 9 and 10 you do not see what is written, even when enlarged.
You say "we made taxonomic revision for P. franchetii depended on evidences from comprehensive morphological features and molecular data", but you don't present any morphometric characterisation or relation between genome and morphologie characteristics; only the normal characterization based on morphologie is presented for the new recombination.